# Social Sustainability?: Exploring the Relationship between Community Experience and Perceptions of the Environment

**Michael R. Cope** *, **Ashley R. Kernan**, **Scott R. Sanders** and **Carol Ward**

Department of Sociology, Brigham Young University, Provo, UT 84602, USA; akernan@byu.edu (A.R.K.); scott_sanders@byu.edu (S.R.S.); carol_ward@byu.edu (C.W.)
* Correspondence: michaelrcope@byu.edu

**Abstract:** This study uses the Rural Utah Community Study (RUCS) to explore how social sustainability shapes a community's approach to environmental sustainability. The results indicate that respondents who feel attached to and are satisfied with their community had a more positive relationship with the natural environment than those who were dissatisfied with their communities. We also find evidence that social ties, measured by the number of people known by their first name in the community, positively influence perceptions of the environment, and that a significant link exists between environmental sustainability and a higher sense of community belonging.

**Keywords:** social sustainability; sustainability; community; environment

## 1. Introduction

The concept of sustainability first appeared in the 1960s [1,2] and later developed into three domains: economic sustainability, environmental sustainability, and social sustainability [2,3]. While the three domains overlap and are interrelated, social sustainability has received much less attention than the others. Among researchers, there is still no agreement about its definition [4–8] or even its precise domain, which has "never actually been agreed upon" [9] (p. 612). While social sustainability is, for some researchers, the ability of a community—or group of people—to achieve any positive result related to sustainability goals, it is more generally agreed to be the "ability of society itself, or its manifestation as local community, to sustain and reproduce itself at an acceptable level of functioning" [2] (p. 293). As a result, models of social sustainability need to consider measures that capture residents' community experience or "the holistic nature of everyday social interaction articulated in a locality, that which is primarily tied to the locality itself" [10] (p. 434).

The purpose of the present study is to assess the extent to which an individual's sense of community, measured by community attachment and satisfaction, influences their connection to the local community. By extending the literature on social sustainability, this study contributes to our understanding of how everyday social interactions relate to the broader goals of general sustainability.

Using data from the Rural Utah Community Study, we asked respondents about their perceptions of the environment and their communities, which produced 1286 responses. First, we review the literature and examine the relationship between community experience and environmental perceptions. We then use ordinary least squares regression to explore these relationships. Finally, we present our findings and their implications.

## 2. Literature Review

### 2.1. Sustainability

Sustainability is a widely used but still poorly developed concept. Recently, to better understand the term, researchers began a more systematic revision of it [6]. For example, Rogge et al. [7] define sustainability as "development that meets the needs of the present

without compromising the ability of future generations to meet their own needs" [7]; that is, the principal goal of sustainability is to use existing resources without negatively influencing their future supply [8]. Because sustainability emphasizes the finiteness of existing resources [9], many countries, including the United States, now recognize sustainability as being essential for governing economic growth [8].

As we have seen, the concept of sustainability now rests on three pillars [11]: environmental sustainability, economic sustainability, and social sustainability [1,4–6,12]. It is asserted that all three are necessary for sustainability to succeed [6], but while this claim is plausible, it is also contentious because, as Purvis et al. [4] suggests, it depends on whether researchers view sustainability as either a single integrated system or as three separate parts [4]. We will begin by considering the parts separately.

### 2.2. Economic Sustainability

Economic sustainability is the best known and understood of the three sustainability domains. Gibbes et al. [6] define it as the preservation and renewal of resources [6], a definition that came about, in part, because of the 2008 Wall Street financial crash that underscored the finiteness of all economic resources [13]. For this reason, economic sustainability now concentrates on sustaining capital resources [14], whether manufacturing, natural, or financial [4] and is a major goal of most economic sustainability policies [4]. It is important to emphasize, then, that while economic sustainability and growth have changed over time, they are now both regarded as growth in human development without harming the economic prospects of future generations [4].

### 2.3. Environmental Sustainability

Environmental sustainability, the best-studied pillar of sustainability, focuses on how we use raw materials to meet human needs and the damage this does to the environment [4]. The rise in interest in environmental sustainability began with the acknowledgement that human economic activity harmed the natural resources, such as clean water, upon which we depend [4]. Hence, environmental sustainability encourages recycling, reusing resources, and mitigating environmental harm [14]. Indeed, there is widespread benefit from environmental sustainability programs, which include energy systems, climate systems, terrestrial systems, carbon and nitrogen systems, aquatic systems, and human settlements and habitats [14], knowing that it is vital to help the environment replenish, repair, and recover from human economic activities [15].

### 2.4. Social Sustainability

Social sustainability, the last of the three sustainability domains, is the least developed [1,16]. This is, in part, due to its recent appearance [16]. What is more, researchers disagree on its definition and domain [16]. By most accounts, social sustainability is a complex topic that crosses several disciplines [1,5,9,17], leading some researchers to define it quite broadly [3]. For example, McKenzie [1] defines it as a "life-enhancing condition within communities," a definition that McCalman [18] generally accepts, but Vallance et al. [3] contend that it means meeting present needs without compromising future generations, which is not so different from the definition of economic sustainability. More generally, Wan and Ng [19] assert that it is a process for promoting both physical and social wellbeing, while Winterton et al. [20] focus on the social benefits of belonging to well-maintained communities. To complicate matters, social sustainability is often measured in different ways, with some researchers concerned more with its effect on local and neighboring communities, rather than on broader communities [1]. It is this need—to better understand social sustainability and how it relates to sustainable development [16]—that this paper seeks to address.

## 2.5. Social Sustainability and Community

Although social sustainability lacks a broadly accepted definition, current research emphasizes its social dimension [16,21], that is, the principle of social equity, which shifts the focus of social sustainability towards equitable access to resources [21]. It is not surprising, then, that an additional focus of social sustainability is the idea of community, which is important "because it provides the context for human thought, relationship, and action" [22]. Models of social sustainability should include processes that capture residents' community experience, as locality is tied to community attachment, thus connecting community attachment to community satisfaction [10,23]. Moreover, community attachment encapsules how residents may feel about their community, whether good or bad [10,23]. It is also focused upon residents' participation in their community, as that has resulted in residents being more invested in their community [24]. When emphasizing community attachment, it is integral to understand that this concept is focused on the local level of community and is more personalized to each resident [10]. When discussing community satisfaction, this concept is focused on the more global sense of the community, making this the bigger picture [10]. It is able to focus on the bigger picture as residents are able to assess their scenarios and can see how that compares to other places throughout society [10]. Community satisfaction is focused on how satisfied residents are within their community [10]. Both community attachment and community satisfaction are overlapping but are different topics within society that can bring a stronger sense of community [10]. As people connect, they nurture a greater sense of belonging, which reinforces the idea of and experiences with community, benefiting both them and where they live [16]. In sum, research indicates that however we define social sustainability, it must include a sense of belonging, attachment, and equity [2,16,21], since they are undeniable benefits worth sustaining [2]. Furthermore, this research contributes to this emerging body of research by demonstrating that when people have equitable access to resources, they are more likely to remain in their communities [2].

## 2.6. Summary and Expectations

As we have seen, much of the literature regarding social sustainability focuses on sustainability in general or on economic and environmental sustainability in particular, with little mention of social sustainability. For this reason, we use a well-known community model [10] to examine the relationships between the community and environmental perceptions. We have chosen this community model because it is well established and will therefore help us to examine the social aspect of communities. As our understanding of social sustainability improves, we will have a better understanding of how it influences the other two pillars of sustainability.

This study, then, asks: "Does having a closer connection and a greater sense of belonging to a community encourage a stronger connection to the environment beyond?" We expect to find that those with a positive community experience will be more connected to the natural environment, both locally and beyond.

## 3. Materials and Methods

### 3.1. Participants and Procedures

The data set came from the Rural Utah Community Study (RUCS). The survey was randomly distributed to 25 rural towns in Utah with populations that varied between 2500 to 5000. Our sampling procedure involved a multi-wave method for mailing the surveys, which meant that before we sent out the surveys, we distributed a letter to potential respondents informing them that a survey would arrive in the next few days. After this initial letter, we sent the survey packet, which included a pre-paid return envelope, a cover letter, $2 compensation, and the 16-page survey.

After receiving the returned surveys, we had a sample of 1286 replies. Accounting for the items returned as undeliverable, the adjusted response rate was approximately 63%.

The survey included questions that asked about community experiences, demographics, and stances towards education.

*3.2. Measures*

3.2.1. Dependent Variable

We used the dependent variable—*the respondents' perceptions of the natural environment*—to determine their connection to the environment beyond their community of residence. This variable was based on twenty-one different statements drawn from the place meaning literature [25,26], and we asked about environment in and around respondents' communities. Specifically, we asked how much they agreed or disagreed with the following: "I am very attached to this area," "I feel this area is part of me," "I strongly identify with this area," "I have pride in my heritage because of this area," "This area is a special place for my family," "Important family memories are tied to this area," "This area is best for the activities I like to do," "I have satisfying experiences when I visit this area," "No other place can compare to this area," "This area is my first choice for outdoor recreation," "I feel that I can really be myself in this area," "Visiting this area says a lot about who I am," "This area contributes to the community's character," "The community's history is defined by this area," "This area helped put the community on the map," "Utah's economy depends on this area," "This area is important for conserving the landscape," "This area is important for providing wildlife habitats," and "This area is important for protecting water quality." The statements were combined and averaged based on respondents' responses to the twenty-one items and then scaled. Each item was coded on a 0–4 measurement, with 0 indicating strong disagreement and 4 indicating strong agreement. The alpha reliability was excellent at 0.94.

3.2.2. Independent Variables

Our first key independent variable, community attachment, was based on two different questions: "How well do you feel that you fit into your community?" and "How much do you have in common with most people in your community?" The responses were averaged and put into a scale. The community attachment scale was coded from 0–6, with 0 indicating that the respondents reported, on average, having a poor fit with their community and nothing in common with other people, and 6 indicating that they reported having a good fit with their community and a lot in common with other people. The alpha reliability score given for this measure was good at 0.81.

Our second key independent variable, community satisfaction, was based on two questions: "How satisfied are you with living in your community?" and "Where would you rank your present community compared with your ideal community?" The responses were put into a scale representing the average for the two questions. The community satisfaction scale was coded on a 0–6 scale, with 0 representing complete dissatisfaction, and 6 indicating strong satisfaction. The alpha reliability was good at 0.85.

Our third key independent variable measures community desirability. Respondents were asked, "Over the past 5 years would you say that, in general, your community has become more desirable, stayed about the same, or become less desirable as a place to live?" The respondent's responses were categorized into three items: 1 = less desirable, 2 = stayed about the same, 3 = more desirable.

Our final key independent variable measures the length of residence in the community. Here we measure length of residence as the proportion of life lived in the community (i.e., the quotient of the number of years lived there divided by age). With rounding, the length of residence ranged from 0 to 1.

3.2.3. Control Variables

In addition to the key independent and dependent variables, the model has a range of control variables. *First name ties* were used to measure how well respondents knew other residents in their community. The question asked, "About what percentage of adults in this

community would you say that you know on a first name basis?" The possible response categories ranged from 0–100%. In the model, it was coded on a 1–4 range, (1 = 0 to 24%, 2 = 25 to 49%, 3 = 50 to 74%, 4 = 75 to 100%). *Children* was used as an additional control variable that asked respondents to indicate how many children they had. The question asked, "How many children, aged 17 or younger, live in your household?" The responses ranged from 0 to 11. The responses were then re-coded to equal the number of children the respondents reported; if they reported more than 7, they were re-coded as 7+.

Further controls were based on the respondent's employment status. The control variables included *retired, full-time, part-time in the labor force,* and *not in the labor force.* The survey asked, "What is your current employment status?" Response categories included "employed for pay in a full time-job," "employed in a part-time job," "unemployed and looking for work," "retired," "homemaker," or "unemployed but not looking for work." Separate dichotomous variables were determined for each category, which included "full-time," "part-time," "in the labor force," "not in the labor force," and "retired." For the *full-time* variable, if respondents indicated they were employed full-time, they were coded as 1, and 0 otherwise. The next variable, *part-time*, was coded similarly to *full-time*. If respondents indicated they were employed *part-time*, they were coded as a 1, and 0 otherwise. The next separate dichotomous variable was *in the labor force*, indicating that the respondent may not be employed but looking for employment. If respondents indicated that they were "unemployed and looking for work," they were coded as a 1, and 0 otherwise. The next variable was *not in the labor force*. If respondents indicated that they were a "homemaker" or "unemployed but not looking for work," they were coded as a 1, 0 otherwise. For the reference category *retired*, respondents who indicated that they were retired were coded to be a 1, and 0 otherwise.

The next control variable was marital status. Our question was "Are you currently married, separated, divorced, widowed, or have you never married?" The *married* variable was re-coded to reference the married category. If respondents indicated they were married, they were coded as a 1, and 0 otherwise.

In addition, *income* was used as a control and was measured by asking: "If you considered all of your earnings, investments, and personal income, about how much was your total family income in 2015?" The answers ranged from $1 to over $150,000 a year. Response category options were 1 = "$1–10,000", 2 = "$10,000–15,000", 3 = "$15,000–20,000", 4 = "$20,000–25,000", 5 = "$25,000–30,000", 6 = "$30,000–35,000", 7 = "$35,000–40,000", 8 = "$40,000–50,000", 9 = "$50,000–60,000", 10 = "$60,000–70,000", 11 = "$70,000–80,000", 12 = "$80,000–90,000", 13 = "$90,000–100,000", 14 = "$100,000–150,000", and 15 = "$150,000+".

The *male* variable accounts for biological sex within our model. Respondents who indicated they were male, were coded as a 1, 0 otherwise. Male was selected as the reference category within this model because more males than females responded to the survey. An additional demographic control included in our model was race. The survey asked, "Which category best describes the ethnic or racial group that you identify yourself with?" The possible response categories were Hispanic/Latino, White, American Indian/Native American, African American/Black, Asian/Pacific Islander, or something else (specify). The variable was coded 1 if white, and 0 otherwise (*white* being the reference group). In addition to controlling for *race*, *age* was also used as a demographic control. The respondents' ages ranged from 18 to 102 years old. This control was used to examine variations in the respondents' answers to items based on their age.

The final three control variables that were used in the model focused on *political affiliation*. Within the model, the political affiliation variable was split into a set of dichotomous variables for which we asked, "When it comes to politics, would you say you are . . . " The possible responses were "liberal," "moderate," or "conservative." The respondents' answers were then coded to indicate their responses. When creating the *liberal* variable, if respondents said they were liberal, they were coded as a 1, and 0 otherwise. For the *moderate* variable, if respondents answered the survey indicating that they were moderate, those answers were coded as a 1, and 0 otherwise. A similar process was used for the

*conservative* variable. However, the *conservative* variable was ultimately left out of the model to indicate that it was the reference group. Descriptive statistics for all variables used in our models are displayed in Table 1.

**Table 1.** Descriptive Statistics.

|  | Mean (Percent) | Standard Deviation |
| --- | --- | --- |
| Dependent Variable |  |  |
| Natural Environment | 2.806 | 0.656 |
| Independent Variables |  |  |
| Community Attachment | 3.979 | 1.227 |
| Community Satisfaction | 4.265 | 1.433 |
| Desirability of Community | 2.067 | 0.690 |
| Proportion of Residence | 53% |  |
| Control Variables |  |  |
| First Name Ties | 2.042 | 0.919 |
| Children | 1.152 | 1.599 |
| Married | 78% |  |
| Employment Status |  |  |
| Retired (Reference Category) |  |  |
| Full Time | 46% |  |
| Part Time | 20% |  |
| In the Labor Force | 0.5% |  |
| Not in the Labor Force | 9% |  |
| Income | 8.415 | 3.864 |
| Male | 57% |  |
| White | 95% |  |
| Age | 50 | 17.650 |
| Political Affiliation |  |  |
| Conservative (Reference Category) |  |  |
| Moderate | 43% |  |
| Liberal | 6.2% |  |

*3.3. Anayltic Stratgey*

To analyze the research question, we used an ordinary least squares (OLS) regression model that predicts a positive association with the natural environment in an individual's community. To account for differential probabilities, bias, and higher levels of non-response rates among segments of the population, we weighted the data based on the distributions for different groups in data taken from zip codes provided by the Rural Utah Community Study (RUCS).

## 4. Results

*4.1. Sample Characteristics*

Table 1 shows the weighted descriptive statistics for the sample. The average age of the sample was 50, and 57% of the sample were male, 95% were white, and 78% were married. On average, respondents reported having one child, 46% had a full-time job, 20% worked a part-time job, 0.5% were in the labor force but not currently working, 9% were not in the labor force, and the remainder were retired—the study's reference category. As for political affiliation, 43% said they were moderate, 6.2% liberal, and the remainder were conservative—our reference category.

*4.2. OLS Regression of Natural Environment*

The OLS regressions, shown in Table 2, predicted whether feeling better about one's community—i.e., more attached, satisfied, and perceiving community as desirable—influenced perceptions of the environment.

**Table 2.** OLS Regression Predicting Environmental Perceptions.

| | Model 1 | | | Model 2 | | | Model 3 | | |
|---|---|---|---|---|---|---|---|---|---|
| | b | | SE | b | | SE | b | | SE |
| Community Attachment | 0.111 | *** | 0.252 | | | | 0.098 | *** | 0.026 |
| Community Satisfaction | 0.183 | *** | 0.021 | | | | 0.181 | *** | 0.023 |
| Desirability of Community | 0.402 | | 0.030 | | | | 0.587 | | 0.068 |
| Proportional Length of Residence | 0.257 | *** | 0.160 | | | | 0.239 | *** | 0.065 |
| *Control Variables* | | | | | | | | | |
| First name ties | | | | 0.197 | *** | 0.026 | 0.054 | * | 0.025 |
| Children | | | | 0.004 | | 0.197 | 0.023 | | 0.016 |
| Married | | | | 0.030 | | 0.649 | 0.033 | | 0.058 |
| Retired (reference) | | | | | | | | | |
| Full Time | | | | 0.028 | | 0.068 | 0.059 | | 0.059 |
| Part Time | | | | 0.091 | | 0.090 | −0.060 | | 0.079 |
| In the Labor Force | | | | 0.437 | | 0.244 | −0.324 | | 0.142 |
| Not in the Labor Force | | | | −0.014 | | 0.127 | 0.004 | | 0.096 |
| Income | | | | 0.008 | | 0.007 | −0.001 | | 0.006 |
| Male | | | | 0.012 | | 0.053 | −0.065 | | 0.047 |
| White | | | | 0.158 | | 0.100 | 0.105 | | 0.092 |
| Age | | | | 0.004 | | 0.002 | 0.000 | | 0.002 |
| Conservative (reference) | | | | | | | | | |
| Moderate | | | | −0.001 | | 0.053 | 0.039 | | 0.043 |
| Liberal | | | | −0.053 | | 0.105 | 0.160 | | 0.092 |
| Intercept | 1.366 | *** | 0.103 | 1.984 | *** | 0.202 | 1.192 | *** | 0.182 |
| $R^2$ | 0.352 | | | | | | 0.119 | | |
| F | 68.661 | | | | | | 6.55 | | |
| P | 0.000 | | | | | | 0.000 | | |

Notes: $n = 915$ * $p < 0.05$ *** $p < 0.001$.

### 4.2.1. Model 1 Results

The relationship between the community experience variables and perceptions of the natural environment are examined in Model 1. Model 1 is a baseline model that focused exclusively on the community experience measures. The results of Model 1 show community satisfaction to have a significant positive association (b = 0.218 $p < 0.001$), and proportional length of residence was also significant and positive (b = 0.439 $p < 0.05$). Thus, within this first model, higher levels of community attachment and proportional length of residence mattered, resulting in greater positive attitudes towards the natural environment. The two other independent variables—*community attachment* and *desirability of community*—were not statistically significant.

### 4.2.2. Model 2 Results

To analyze the relationship between measures of community and perceptions of the natural environment, Model 2 focused exclusively on using the systematic models of community. The results of this model indicated that only first name ties were significant (b = 0.218 $p < 0.001$), which suggests that greater community ties can have a positive effect on perceptions of the natural environment. The other variables in Model 2 were not significant in terms of positive perceptions of the natural environment.

### 4.2.3. Model 3 Results

Next, to examine the relationship between perceptions of the natural environment and an individual's community experience, we integrated both the community experience variables and the systematic models of community in Model 3. The results indicate that three of the community experience variables—*community attachment, community satisfaction, and proportional length of residence*—were all statistically significant at the $p < 0.001$ level. Community attachment (b = 0.098 $p < 0.001$) was positively associated with perceptions

of the natural environment, and community satisfaction (b = 0.181 *p* < 0.001) was also positively associated with perceptions of the natural environment. The last statistically significant independent variable was the proportional length of residence, which became statistically more significant than in Model 1 (b = 0.239 *p* < 0.001). This was also positively associated with perceptions of the natural environment. The fourth independent variable—*the desirability of the community*—was not statistically significant, and all but one control variable was not statistically significant (*p* > 0.05). The control variable that did have significance was *first name ties*, which showed a positive association (b = 0.054 *p* < 0.05). Compared to Model 1, the relationship between *first name ties* and the independent variable indicated a positive relationship, though it was less significant.

In addition, Model 3 reveals some important associations. All but one of the independent variables were statistically significant (*p* < 0.001). Among the independent variables of community attachment, community satisfaction, and proportional length of residence, the highest coefficient was the proportional length of residence, followed by community satisfaction. However, regardless of the coefficient numbers, the three independent variables and one control variable were all statistically significant, indicating a positive relationship. Thus, community attachment, community satisfaction, and proportional length of residence matter within this relationship. In addition to the independent variables, one control, *first name ties*, was also significant and matters in the relationship.

## 5. Discussion

This paper seeks to contribute to the concept of social sustainability by examining whether a close connection and greater sense of belonging to a community encourage a willingness to preserve the environment.

By examining the existing research, we found that the concept of social sustainability remains underdeveloped [6,7,16,21]. However, our discussion of relevant literature suggests that social sustainability includes multiple elements [2,16,21], such as community attachment, community satisfaction, the desirability of community, and the proportional length of residence [10,22,23]. Using regression analysis, we tested how these might influence community residents' perceptions of the local natural environment.

Our statistical analysis asked, "How does feeling more positive about your community—i.e., feeling more attached to, more satisfied with, and that your community is more desirable—affect your views of the natural environment?" Our results indicate that community attachment, community satisfaction, and proportional length of residence all mattered, which suggests that stronger community ties can lead to a positive increase in how respondents feel about their local environment. That is, as people feel more attached to, satisfied with, and live longer in a community they are more willing to protect the local environment. This shows that a higher attachment to and greater satisfaction with a community are significant factors in the cultivation of sustainability. It is also important to note that the first name ties were statistically significant, which indicates that the larger percentage of residents who know others by their first names contributes to stronger community and environmental ties. These findings are consistent with previous research that suggests social sustainability is multifaceted [6,7,16,21]. One unexpected finding, however, was that the desirability of the community within the regression was not significant and indicates that respondents' feelings of desirability of their community did not matter for social and environmental sustainability.

This study illuminates key community relationships and provides new evidence for the relationship between social sustainability, represented by key elements of community, and its effects on environmental sustainability. Our findings can be used as a framework for future researchers to better understand the theoretical aspects of social sustainability and how it can influence residents' perceptions of their local environment. The findings also show that the ways people feel towards their communities influence how they perceive their environments. According to our research, without social sustainability, environmental sustainability will not be as effective, if it is effective at all.

The present study has several limitations. For example, we considered only rural residents of Utah, USA; therefore, our findings may not reflect the community experience of residents elsewhere. Future researchers should examine the patterns in other populations to further our understanding of social and environmental sustainability. Another limitation is that our sample is rather homogenous. An example of this is shown in Table 1, which indicated that 95% of those who took the survey were white, 57% were male, and 78% were married. Although we weighted the data to account for overrepresented groups, overrepresentation can still occur with so many respondents indicating similar demographic characteristics. Future researchers might consider a more diverse sample to help account for these over-representations of particular groups. Even so, we believe the present study provides meaningful results that help to clarify the relationship between the community and environmental sustainability. As the idea of social sustainability is critically underdeveloped, this study helps researchers and other interested parties to understand its implications.

## 6. Conclusions

Social sustainability is complex and understudied. Our findings reveal how important it is for people to feel attached to and satisfied with their communities since they are then more likely to take better care of their communities and have sustainable neighborhoods. To that end, social sustainability should be seen as a pivotal factor for healthy communities. Our research offers a glimpse into how social sustainability within a community relates to environmental sustainability. Furthermore, this study asserts that community attachment cultivates respect and care for the local environment. To that end, future research should be designed to address the sustainability of not only the physical—built—aspects of a local community, but also the nonphysical—social—dimensions within a community. Moreover, policy should seek to recognize community residents as emplaced allies who will act responsibly as stewards of the local environment while, at the same time, securing their livelihoods and social practices.

**Author Contributions:** Conceptualization, M.R.C.; Data curation, M.R.C., S.R.S. and C.W.; Formal analysis, M.R.C. and A.R.K.; Funding acquisition, M.R.C., S.R.S. and C.W.; Methodology, M.R.C.; Project administration, M.R.C. and C.W.; Supervision, M.R.C.; Writing—original draft, M.R.C., A.R.K. and S.R.S.; Writing—review & editing, A.R.K. and S.R.S. All authors have read and agreed to the published version of the manuscript.

**Funding:** The author(s) disclosed receipt of the following financial support for the research, authorship, and/or publication of this article: Data collection efforts were funded in part by two sources internal to Brigham Young University: The Charles Redd Center for Western Studies, and the College of Family, Home, and Social Sciences.

**Institutional Review Board Statement:** This study was conducted according to the guidelines of the Declaration of Helsinki, and approved by the Institutional Review Board of Brigham Young University (protocol code: E16349 approved on 15 September 2015).

**Informed Consent Statement:** All persons gave their informed consent prior to their inclusion in this study.

**Data Availability Statement:** The datasets generated during and/or analyzed during the current study are available from the corresponding author on reasonable request. The data are not publicly available due to the inclusion of information that could compromise the privacy of the research participants.

**Acknowledgments:** The authors thank Jorden E. Jackson, Paige N. Park, and the students in BYU Communities Studies Lab for help during the data collection and curation phases of the project.

**Conflicts of Interest:** The authors declare no conflict of interest.

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
