# Peer review of "Social Sustainability?: Exploring the Relationship between Community Experience and Perceptions of the Environment"

_sustainability, doi:10.3390/su14031935_

Round 1

Reviewer 1 Report

Summary

The aim of the paper is to explore the concept of social sustainability by investigating the relationship  between individuals’ perception of the environment where they live and the attachment and satisfaction with their community using survey data from the Rural Utah Community Study.

Main comments

Exploring the concept of social sustainability is a worthwhile endeavor as a consensus on its definition and how to measure or assess it has not been reached yet. However, the paper does not provide neither clear theoretical framework nor convincing empirical evidence.

First part-literature review

The number of alternative definitions of social sustainability is extremely large. Most of them are focused on the idea of future generations having access to the same/greater level of social resources as the current generation, where the definition of social resources or minimal social requirements is still to be defined (it typically encompasses cultural heritage, human rights, labour rights, social justice,  livability, health equity).

It is often (but not always) the case that the definitions make explicit reference to the physical dimension (infrastructures) that support social and cultural lives and allow citizen engagement and development.

The literature review proposed does not provide a complete and updated review of the existing definitions. More detail should be provided on what are the main differences between them, what is the main challenge in finding a common definition etc. 

A strong focus should be placed on the concept of community, its definition and how it is interrelated with social sustainability.

Furthermore, given the subsequent empirical attempt, some space should be dedicated to look at which criteria are typically used to measure social sustainability (make also reference to the sustainable development goals) and which variables have been used by the empirical literature so far.

The authors make reference to a community model by Brown et al. But a more clear description of the model and why it has been chosen should be provided

My suggestion is to limit the description of the other two sustainable pillars to a paragraph at most and then focus entirely on social sustainability.

Second part- empirical analysis

Research question

The authors want to investigate if an individual’s sense of community influences its perception on the environment beyond the community. This research question is not clear.

What do the authors mean by “environment beyond the community” (sometimes called “natural environment”, “surrounding environment)?  Is it intended as the social infrastructures (schools, museums, parks, various amenities), or something else? And what is it intended with “beyond” community? 

The authors, here and there, claim also that this can inform on how individual perceive community environmental policies towards environmental sustainability and if individuals are more willing to protect the environment.  I believe this is not addressed in the paper, or this is not clear.

The author should be clearer in explaining their research question and how is it related to the definition of social sustainability and how it contributes to the existing literature.

Empirical analysis

For what concerns the empirical analysis, more information should be provided on:

  1.  the survey. For example: in what year was it administered? Are the considered rural towns recent settlements or old ones, do they have particular cultural or historical characteristics? The sense of identity or         community can be driven by such kind of characteristics. Maybe the authors could provide  a map of the rural towns, or a description of the area considered?
    2.    The sample: how many observations are there for each town? Are the towns very similar from a social/cultural standpoint or is there heterogeneity?
    3.    The variables. It is not clear what type of answers were available to respondents to the questions on communities. The authors should provide standard deviation also for the dummy variables.
    4.    The authors should explain why some variables (such as employment status or gender or political affiliation or income) are expected to have an impact on the independent variable and how.

Regarding the model, the independent variable seems to include very different opinions asked to the respondent regarding the environment (that I intend as the physical area where the respondent resides), that can be divided in three types: 1. Feeling of cultural and social identity and heritage 2. Access to services 3. Environmental issues. These are very different opinion that capture different preferences. Probably, only some of them are relevant for the analysis or, if all, they should be treated separately.

As mentioned, it is not clear why some dependent variable could be relevant in the model, while some others might have been omitted (e.g. level of availability of social/cultural services, macro-indicators of quality of life). Clearly, instead, it is very relevant to include a categorical variable that associated the observations to the town where they belong, so that some of the macro omitted variables can be captured (maybe this is what the authors wanted to do by weighting observations, but that was not clear)

Regarding desirability of community, it should be very much correlated to the independent variable as they are similar questions? Something is not clear.

On the models, only Model 3 is relevant to the analysis, the other Models can be omitted.

There could be a big problem with reverse causality in the model: if I have a good perception of the environment in which I live in, I could also feel closer to my community.

Other comments:

There are many typos in the manuscript that should be corrected (see line 37, 43, 46, 72,79…).

The description of the independent variables (section 3.2.2) should be cut and more space should be dedicated to how these variable are relevant to the analysis. 

Author Response

Reviewer 1

Main comments

Exploring the concept of social sustainability is a worthwhile endeavor as a consensus on its definition and how to measure or assess it has not been reached yet. However, the paper does not provide neither clear theoretical framework nor convincing empirical evidence.

First part-literature review

The number of alternative definitions of social sustainability is extremely large. Most of them are focused on the idea of future generations having access to the same/greater level of social resources as the current generation, where the definition of social resources or minimal social requirements is still to be defined (it typically encompasses cultural heritage, human rights, labour rights, social justice, livability, health equity).

It is often (but not always) the case that the definitions make explicit reference to the physical dimension (infrastructures) that support social and cultural lives and allow citizen engagement and development.

The literature review proposed does not provide a complete and updated review of the existing definitions. More detail should be provided on what are the main differences between them, what is the main challenge in finding a common definition etc. 

A strong focus should be placed on the concept of community, its definition and how it is interrelated with social sustainability.

Furthermore, given the subsequent empirical attempt, some space should be dedicated to look at which criteria are typically used to measure social sustainability (make also reference to the sustainable development goals) and which variables have been used by the empirical literature so far.

The authors make reference to a community model by Brown et al. But a more clear description of the model and why it has been chosen should be provided

My suggestion is to limit the description of the other two sustainable pillars to a paragraph at most and then focus entirely on social sustainability.

Authors’ Response: Thank you for these comments. Based up your feedback, the Introduction and Literature Review were completely rewritten to address the concerns detailed in your review.

Second part- empirical analysis

Research question

The authors want to investigate if an individual’s sense of community influences its perception on the environment beyond the community. This research question is not clear.

What do the authors mean by “environment beyond the community” (sometimes called “natural environment”, “surrounding environment)?  Is it intended as the social infrastructures (schools, museums, parks, various amenities), or something else? And what is it intended with “beyond” community? 

The authors, here and there, claim also that this can inform on how individual perceive community environmental policies towards environmental sustainability and if individuals are more willing to protect the environment.  I believe this is not addressed in the paper, or this is not clear.

The author should be clearer in explaining their research question and how is it related to the definition of social sustainability and how it contributes to the existing literature.

Authors’ Response: Based on your comments, Sections 2.5 and 2.6 were redrafted to clarify references to community, the research question and contributions of this research to the literature. We appreciate your feedback and believe the paper is significantly strong after our revisions.

Empirical analysis

For what concerns the empirical analysis, more information should be provided on:

  1. The survey. For example: in what year was it administered? Are the considered rural towns recent settlements or old ones, do they have particular cultural or historical characteristics? The sense of identity or community can be driven by such kind of characteristics. Maybe the authors could provide a map of the rural towns, or a description of the area considered?
    The sample: how many observations are there for each town? Are the towns very similar from a social/cultural standpoint or is there heterogeneity?
    3.    The variables. It is not clear what type of answers were available to respondents to the questions on communities. The authors should provide standard deviation also for the dummy variables.
    4.    The authors should explain why some variables (such as employment status or gender or political affiliation or income) are expected to have an impact on the independent variable and how.

Regarding the model, the independent variable seems to include very different opinions asked to the respondent regarding the environment (that I intend as the physical area where the respondent resides), that can be divided in three types: 1. Feeling of cultural and social identity and heritage 2. Access to services 3. Environmental issues. These are very different opinion that capture different preferences. Probably, only some of them are relevant for the analysis or, if all, they should be treated separately.

As mentioned, it is not clear why some dependent variable could be relevant in the model, while some others might have been omitted (e.g. level of availability of social/cultural services, macro-indicators of quality of life). Clearly, instead, it is very relevant to include a categorical variable that associated the observations to the town where they belong, so that some of the macro omitted variables can be captured (maybe this is what the authors wanted to do by weighting observations, but that was not clear)

Regarding desirability of community, it should be very much correlated to the independent variable as they are similar questions? Something is not clear.

Authors’ response: Sections 3.2.1 - 3.2.3 have been edited to include information that addresses the reviewer’s questions related to the survey, the sample, as the variables including control variables. In section 3.2.2, we have edited the information to clarify our focus on the independent variables which are related to community attachment, satisfaction and desirability as well as length of residence for this model.

On the models, only Model 3 is relevant to the analysis, the other Models can be omitted.

There could be a big problem with reverse causality in the model: if I have a good perception of the environment in which I live in, I could also feel closer to my community.

Authors’ response: Reverse causality is a concern. As a result, we ensured the language used in our revisions discusses correlation and how that theoretically contributes to the academic literature. Elucidating the causal mechanism is beyond the score of this research.

Other comments:

There are many typos in the manuscript that should be corrected (see line 37, 43, 46, 72,79…).

The description of the independent variables (section 3.2.2) should be cut and more space should be dedicated to how these variable are relevant to the analysis. 

Authors’ response: Thank you for these comments. We have corrected the typos. The rewritten and heavily edited sections addressed the concern about the independent variables and their relevance to the analysis.

Reviewer 2 Report

  1. Abstract:
    • Please rewrite the aim of the study defined in the abstract (“This article aims to explore and uncover notions of social sustainability”, which is too general, not consistent with the aim statement in the introduction section, and does not represent the gist of the article (REQUIRED).
    • Please consider supplementing the abstract with a short description of ‘added value’ of the paper for theory and practice (OPTIONAL).
    • Highlights (i.e. “a short collection of bullet points that convey the core findings of the article”) or a graphical abstract are not mandatory. Nevertheless, I strongly recommend to supplement the paper with them as they increase attractiveness of the paper for potential readers and increase the number of reads on-line (OPTIONAL).
  2. Introduction section:
    • The introduction section is well structured and written. No additional actions required.
  3. Literature review section:
    • The literature review section is well structured and written. No additional actions required.
  4. Methodology section
    • It is stated that the dependent variable consists of 21 items, while only 19 of them are listed. Please provide 2 missing items (REQUIRED).
    • Please check and validate the logic in Lines 306-309 (REQUIRED).
  5. Results presentation section
    • I strongly recommend to add the graphical representation of the tested model (figure) to make the findings more readable (REQUIRED).
    • When comparing the numbers (b, p) in Sections 4.2.1 and 4.2.2 with data in Table 2, differences are noticed. Please remove these ambiguities (REQUIRED).
  6. Discussion and conclusion section
    • In Section 2.6, the Authors define the following research question: “Does having a closer connection and greater sense of belonging to one’s community encourage more a stronger connection to the environment beyond the community?” (Lines 172-174). Then in the discussion section they make attempts to provide an answer to another question: “How does feeling better about your community, i.e., feeling more attached, satisfied, and more desirable about your community, affect your one’s views of the natural environment?” (Lines 401- 403). I strongly recommend to define one research question, to make Authors intentions unambiguous to the Readers (REQUIRED).
    • Compare and contrast the findings of your study with other publications and/or establish links between the findings of your study and theory. This should be the main point of the discussion section and would increase quality of discussion. Try to provide responses to questions: WHY? SO WHAT? (REQUIRED).

Author Response

Reviewer 2

  1. Abstract:
    • Please rewrite the aim of the study defined in the abstract (“This article aims to explore and uncover notions of social sustainability”, which is too general, not consistent with the aim statement in the introduction section, and does not represent the gist of the article (REQUIRED).
    • Please consider supplementing the abstract with a short description of ‘added value’ of the paper for theory and practice (OPTIONAL).
    • Highlights (i.e. “a short collection of bullet points that convey the core findings of the article”) or a graphical abstract are not mandatory. Nevertheless, I strongly recommend to supplement the paper with them as they increase attractiveness of the paper for potential readers and increase the number of reads on-line (OPTIONAL).

Authors’ response: Thank for you for these comments. The Abstract has been rewritten to clarify the aim of the study.

  1. Introduction section:
    • The introduction section is well structured and written. No additional actions required.

Authors’ response: Thank you for your comment.

  1. Literature review section:
    • The literature review section is well structured and written. No additional actions required.

Authors’ response: Thank you for your comment.

  1. Methodology section
    • It is stated that the dependent variable consists of 21 items, while only 19 of them are listed. Please provide 2 missing items (REQUIRED).
    • Please check and validate the logic in Lines 306-309 (REQUIRED).

Authors’ response: Thank you for your comment. Sections 3.2.1 - 3.2.3 have been edited to include information that addresses the reviewer’s concerns about the variables and logic.

  1. Results presentation section
    • I strongly recommend to add the graphical representation of the tested model (figure) to make the findings more readable (REQUIRED).
    • When comparing the numbers (b, p) in Sections 4.2.1 and 4.2.2 with data in Table 2, differences are noticed. Please remove these ambiguities (REQUIRED).

Authors’ response: Sections 4.2.1 and 4.2.2 were carefully edited to ensure the data in the text matches the data.

  1. Discussion and conclusion section
    • In Section 2.6, the Authors define the following research question: “Does having a closer connection and greater sense of belonging to one’s community encourage more a stronger connection to the environment beyond the community?” (Lines 172-174). Then in the discussion section they make attempts to provide an answer to another question: “How does feeling better about your community, i.e., feeling more attached, satisfied, and more desirable about your community, affect your one’s views of the natural environment?” (Lines 401- 403). I strongly recommend to define one research question, to make Authors intentions unambiguous to the Readers (REQUIRED).
    • Compare and contrast the findings of your study with other publications and/or establish links between the findings of your study and theory. This should be the main point of the discussion section and would increase quality of discussion. Try to provide responses to questions: WHY? SO WHAT? (REQUIRED).

Authors’ response: Thank you for your comments. We have edited and rewritten the results sections, 4.1, 4.2, 4.2.1 – 4.2.3 as well as Section 5 to address these comments and extend our discussion of the results and their relevance to theory and related studies.

Reviewer 3 Report

The main ad very important issue of this manuscript is that the authors mix up the two notions: sustainability and sustainable development (see eg. lines 20-38, 51-128). That is clear from their review of the three pillars (economic, environmental, social) of sustainable development that they present as the three pillars of sustainability. 

For example, the authors say that '[t]o some, social sustainability is conceptualized as the ability of a given community—or group of people—to achieve any positive result related to a given sustainability goal.' (lines 29-30). Do they mean one of the UN Sustainable Development Goals (SDGs) or not? If so, they must state it clearly and also make references to the UN SDGs.

It is all the more confusing when the authors cite certain Rogge et al. [footnote 7] who allegedly defined sustainability as the “development that meets the needs of the present without compromising the ability of future generations to meet their own needs” (lines 55-56). Well, that's not the definition of sustainability that someone gave in 2018 as the authors suggest, but the international nomenclature definition of sustainable development given yet in 1987 in 'Our Common Future', also known as the Brundtland Report, published by the United Nations through the Oxford University Press. Brundtland Report was published in recognition of Gro Harlem Brundtland's, former Norwegian Prime Minister, role as Chair of the World Commission on Environment and Development.

Last, but not least, the three pillars (economic, environmental, social) that the authors discuss in their literature review belong to sustainable development and not to sustainability. This three-tier model of sustainable development also has its sources, today not only secondary (academic literature), but also primary (various international conventions and other instruments).

All in all, the authors must update and refine the whole literature review as follows:
- they give the nomenclature definition of sustainable development with clear and correct references to the sources of the definition
- they give their working definition of sustainability
- if the latter is similar to the former, they say it clearly
- if not, they explain why and how the definition/ concept of sustainability may (and maybe even should) be different from sustainable development 
- when they speak about three pillars (economic, environmental, social) of sustainable development, they also provide concrete references to the three-tier model of sustainable development (not sustainability!)

Author Response

Reviewer 3

The main and very important issue of this manuscript is that the authors mix up the two notions: sustainability and sustainable development (see e.g., lines 20-38, 51-128). That is clear from their review of the three pillars (economic, environmental, social) of sustainable development that they present as the three pillars of sustainability. 

For example, the authors say that '[t]o some, social sustainability is conceptualized as the ability of a given community—or group of people—to achieve any positive result related to a given sustainability goal.' (lines 29-30). Do they mean one of the UN Sustainable Development Goals (SDGs) or not? If so, they must state it clearly and also make references to the UN SDGs.

It is all the more confusing when the authors cite certain Rogge et al. [footnote 7] who allegedly defined sustainability as the “development that meets the needs of the present without compromising the ability of future generations to meet their own needs” (lines 55-56). Well, that's not the definition of sustainability that someone gave in 2018, as the authors suggest, but the international nomenclature definition of sustainable development given yet in 1987 in 'Our Common Future', also known as the Brundtland Report, published by the United Nations through the Oxford University Press. Brundtland Report was published in recognition of Gro Harlem Brundtland's, former Norwegian Prime Minister, role as Chair of the World Commission on Environment and Development.

Last, but not least, the three pillars (economic, environmental, social) that the authors discuss in their literature review belong to sustainable development and not to sustainability. This three-tier model of sustainable development also has its sources, today not only secondary (academic literature), but also primary (various international conventions and other instruments).

All in all, the authors must update and refine the whole literature review as follows:
- they give the nomenclature definition of sustainable development with clear and correct references to the sources of the definition
- they give their working definition of sustainability
- if the latter is similar to the former, they say it clearly
- if not, they explain why and how the definition/ concept of sustainability may (and maybe even should) be different from sustainable development 
- when they speak about three pillars (economic, environmental, social) of sustainable development, they also provide concrete references to the three-tier model of sustainable development (not sustainability!)

Authors’ response: Thank you for these comments. We have rewritten the paper to clarify our purpose to establish a unifying concept of social sustainability by examining whether a close connection and greater sense of belonging to a community encourage a willingness to preserve the environment.

Round 2

Reviewer 1 Report

The current revised version of the document satisfactorily addresses all comments made on the previous version.

Author Response

Main comments

The current revised version of the document satisfactorily addresses all comments made on the previous version.

Authors’ Response: Thank you for these (and your previous) comments. We believe that the revisions we have made based on these comments have allowed us to develop a much-improved manuscript. Thank you.